# Engineering Terpene Production Pathways in *Methylobacterium extorquens* AM1

**DOI:** 10.3390/microorganisms12030500

**Published:** 2024-02-29

**Authors:** Allison Hurt, Jacob D. Bibik, Norma Cecilia Martinez-Gomez, Björn Hamberger

**Affiliations:** 1Department on Microbiology and Molecular Genetics, Michigan State University, East Lansing, MI 48824, USA; 2Department of Biochemistry and Molecular Biology, Michigan State University, East Lansing, MI 48824, USA; jacob.bibik@gmail.com (J.D.B.); hamberge@msu.edu (B.H.); 3DOE Great Lakes Bioenergy Research Center, Michigan State University, East Lansing, MI 48824, USA; 4Department of Plant and Microbial Biology, University of California Berkeley, Berkeley, CA 94709, USA; cecimartinez@berkeley.edu

**Keywords:** metabolism, terpenes, patchoulol, casbene, methylotrophs, inducible expression, lanthanide

## Abstract

Terpenes are diverse specialized metabolites naturally found within plants and have important roles in inter-species communication, adaptation and interaction with the environment. Their industrial applications span a broad range, including fragrances, flavors, cosmetics, natural colorants to agrochemicals and therapeutics, yet formal chemical synthesis is economically challenging due to structural complexities. Engineering terpene biosynthesis could represent an alternative in microbial biotechnological workhorses, such as *Saccharomyces cerevisiae* or *Escherichi coli*, utilizing sugars or complex media as feedstocks. Host species that metabolize renewable and affordable carbon sources may offer unique sustainable biotechnological alternatives. Methylotrophs are bacteria with the capacity to utilize one-carbon feedstocks, such as methanol or formate. They colonize the phyllosphere (above-ground area) of plants, and many accumulate abundant carotenoid pigments. Methylotrophs have the capacity to take up and use a subset of the rare earth elements known as lanthanides. These metals can enhance one-carbon (methylotrophic) metabolism. Here, we investigated whether manipulating the metabolism enables and enhances terpene production. A carotenoid-deficient mutant potentially liberates carbon, which may contribute to bioproduct accumulation. To test this hypothesis, terpene-producing bacterial strains regulated by two distinct promoters were generated. Wildtype *Methylobacterium extorquens*, ∆*Meta1_3665*, a methylotrophic mutant lacking the carotenoid pathway, and an *E. coli* strain were transformed with an exogenous terpene pathway and grown both in the presence and absence of lanthanides. The extraction, and the comparison of analytical profiles, provided evidence that engineered cultured *M. extorquens* under control of a native, inducible methylotrophic promoter can yield the sesquiterpene patchoulol when supplemented with lanthanide. In contrast, using a moderate-strength constitutive promoter failed to give production. We demonstrated colonization of the phyllosphere with the engineered strains, supporting the future engineering of selected species of the plant microbiome and with promising implications for the synthetic biology of small molecules.

## 1. Introduction

Plants produce specialized metabolites including a broad spectrum of terpenoids that are significant in defense mechanisms, communication and adaptation [1]. Terpenoids also represent industrially relevant bioproducts with a wide range of existing and potentially future renewable applications, including perfumes, pharmaceuticals and food supplements [2]. Typical terpene biosynthetic pathways proceed via the universal 5-carbon precursors isopentenyl diphosphate (IDP) and dimethylallyl diphosphate (DMADP), formed in plants by the cytosolic mevalonate (MVA) and plastidial methylerythritol phosphate (MEP) pathways, with the latter route also found in bacteria. IDP and DMADP are condensed into acyclic isoprene diphosphates of varying lengths in multiples of 5-carbons. Subsequently, terpene synthases can cyclize the intermediates into complex scaffolds with the characteristic monoterpenes (C_10_), sesquiterpenes (C_15_), diterpenes (C_20_) or carotenes/tetraterpenes (C_40_), as examples. The extraction of terpenes from their native species for industrial applications poses environmental and economic issues with harvesting, limitations in the amount and purification from mixtures of related but unwanted products [3]. Economically viable chemical synthesis is challenging due to its structural complexity [4]. For these reasons, the bioengineering of terpene production through biotechnology is rising in interest for its sustainable production [5,6,7]. 

The early intermediate pathways are crucial for terpene synthesis and may enable the rewiring of non-plant organisms to host exogenous terpene production. In microbial hosts, the levels of intermediates have been increased through the installation of strong native pathways and engineered alternative routes, as well as the removal or suppression of competing pathways, including sterols/triterpenes (C_30_) or carotenoids [8,9,10,11]. However, these manipulations may require multiple plasmids to supply the terpene synthase and upregulate the precursors and can represent a burden on the system [12]. These challenges, coupled with the need for glucose or complex feedstocks, raise an opportunity for terpene production in genetically tractable organisms with unique carbon metabolism abilities, such as *Methylobacterium extorquens* AM1 [13].

Methylotrophs are bacteria unique in their ability to utilize one-carbon compounds including low-cost feedstocks such as methanol or formate. They colonize a spectrum of environments but are exceptionally abundant in the phyllosphere (above-ground area of plants), where they utilize methanol released by cell wall catabolism [14]. Many methylotrophs accumulate carotenoids, giving the microbes a visibly pink hue. Methylotrophs are unique in their uptake and use of rare earth elements known as lanthanides. These metals can alter and enhance one-carbon (methylotrophic) metabolism. *M. extorquens* AM1 modulates the production of different methanol dehydrogenase (MDH) enzymes (XoxF- and ExaF) depending on the concentration of lanthanides present in the media, allowing a robust metabolic flexibility with the changing of substrate concentrations [15]. Additionally, lanthanides visually intensify the color of cultured isolates and have proven to be essential for growth with some species [16]. Using the MEP pathway, methylotrophs generate abundant terpene precursors for carotenoid synthesis. A previous study isolated the carotenoid-deficient strain CM502, which lacked proposed diapolycopene oxidase activity (META1_3665, GenBank accession AY331188.1) [17]. It is possible that in CM502, the pool of precursors may be liberated and accessible by exogenous pathways. A methylotrophic strain with intact native carotenoid biosynthesis has also been engineered with an exogenous terpene pathway, yet with low resulting yields [13]. The impact of lanthanides on the carbon flow through the carotenoid pathway has not been investigated and presents an additional opportunity to test for enhanced terpene accumulation.

Here, we focused on two terpene products, patchoulol (C_15_ sesquiterpene) and casbene (C_20_ diterpene). Patchoulol is a sesquiterpene alcohol relevant in the perfume and cosmetic industry and known for its earthy aroma. It is naturally formed by cyclization of farnesyl diphosphate (FDP) catalyzed by a sesquiterpene synthase and is the main component (30–40%) of patchouli oil extracted from the *Pogostemon cablin* plant [1,18,19]. Industrialization of the compound has increased its demand, with prices ranging from $30–$200 per kg [2]. The range in price can be accounted for by the limitations and biological variation in cultured plants. Casbene is a diterpene with antimicrobial effects, formed through a one-step cyclization of geranylgeranyl diphosphate (GGDP) [20]. Casbene synthase, *DgTPS1*, is natively found in the *Daphne genkwa* plant [21]. The scaffold can be further modified to afford important precursors in drug discovery [22]. 

The CM502 mutant arrests the carotenoid route, potentially liberating carbon for the biosynthesis of novel products. It was reported, after metabolic modifications, that CM502 synthesized more of the sesquiterpenoid target α-humulene than the wildtype strain [13]. The introduction of the patchoulol synthase under a native promoter may take advantage of the FDP pool. However, there may be an increase of the products that follow, such as GGDP, allowing for the generation of the diterpene casbene. Given there are two potential metabolite branch points, methylotrophic expression may have the opportunity to increase production, lower cost, and reduce the environmental impacts of terpene synthesis. 

This study aims to increase the knowledge of terpene engineering specifically within the native and mutant background of *M. extorquens* AM1. To test if lanthanides can be used to increase terpene synthesis in methylotrophs, we supplemented engineered and cultured strains with the rare earth metal LaCl_3_. To investigate if the engineered strains retain the capacity to colonize plants and persist in the phyllosphere, we recovered *M. extorquens* from leaves four days after inoculation. 

## 2. Materials and Methods

### 2.1. Strains, Media, and Growth Conditions

MP minimal salts, CoCl_2_, succinic acid, methanol, and chloramphenicol were purchased from Sigma (Sigma Aldrich, St. Louis, MO, USA), Streptomycin, LB, and TB were purchased from VWR (VWR International, LLC, Radnor, PA, USA). *M. extorquens* AM1 wildtype and mutant CM502 were used in the study [17] and grown on MP minimal salts agar plates (1.5% *w*/*v*) with a CoCl_2_ concentration of 2.0 μM and 30 mM succinic acid as a carbon source [23]. For liquid growth, MP minimal salts media with 2.0 μM CoCl_2_ and 125 mM methanol as a carbon source was used [23]. Cultures were incubated at 28.5 °C with the shaking of liquid cultures at 175 rpm. For methylotrophic engineered strains, 12.5 μM CoCl_2_ was used in MP media with the antibiotic tetracycline hydrochloride (Sigma Aldrich, St. Louis, MO, USA) for selection in a final concentration of 10 μg/mL. The *E. coli* expression system required tetracycline and streptomycin with a final concentration of 50 μg/mL and chloramphenicol, with a final concentration of 34 μg/mL in the two-plasmid system and 17 μg/mL in the three-plasmid system. *E. coli* strains were grown on LB plates (1.5% *w*/*v*) for solid media. Liquid cultures were in LB or TB media with appropriate antibiotics. All cultures were wrapped in tin foil or kept in the dark to prevent tetracycline degradation. 

### 2.2. Plasmid Generation and Electroporation

Plasmids were generated through PCR amplification and In-Fusion^®^ (Takara Bio USA, Inc., Ann Arbor, MI, USA) cloning. All primers and plasmids used in this study can be found in Table 1 and Table 2, respectively. A modified *Tac* promoter *mTac* [24,25] was chosen for expression in both *E. coli* and methylotrophs to enable a comparison of the impact of lanthanides on carotenoid production. pAP5 [26] was used as a backbone and amplified with primers 1 and 2 to generate a vector for gene and promoter insertion. Patchoulol synthase *PcPatS* (accession number: AY508730) [27] was amplified using primers 3 and 4. The *mTac* promoter was synthesized and amplified with primers 5 and 6. When appropriate, overhangs complementary to the gene or backbone next to the insertion site were generated on primers for In-Fusion^®^ HD Cloning Plus (Takara Bio) cloning, and plasmids were transformed into Stellar Competent Cells (Takara Bio). In the pAP5 backbone, the *PcPatS* gene was inserted with the *mTac* promoter, resulting in the pAH1 vector (Table 2). Appropriate constructs were selected on LB plus tetracycline hydrochloride plates and confirmed by PCR and through Sanger sequencing (Psomagen, Rockville, MD, USA).

To investigate the production by a native, lanthanide-inducible promoter, the gene encoding the patchoulol synthase was inserted into a pES503, a plasmid containing the *xox1* methylotrophic promoter [26], to generate pAH2 (Table 2). Analogously, the gene encoding casbene synthase *DgTPS1* (accession number: MZ485349.1) [21] was amplified using the 9 and 10 primers, resulting in pAH3 (Table 2). A construct containing instead the *mTac* promoter was generated by using primers 11 and 10, resulting in pAH4 with casbene synthase (Table 2). Constructs were verified by Sanger sequencing. For the *E. coli* expression system, previously established plasmids for increasing the precursors needed, pIRS and pGG, were utilized [21,28]. An overview of the MEP pathway and vectors is given in Figure 1.

Competent cells of both AM1 and CM502 were generated using a previously reported method [29]. Overnight cultures of both strains were started from plates and used to inoculate 50 mL flasks of MP media with methanol. The following day, cultures were transferred into sterile 50 mL falcon tubes and centrifuged at 4000× *g* for 15 min at 4 °C. The media was discarded and chilled sterile ddH_2_O was used to resuspend the pellet. Another round of centrifugation and wash was performed. After the final spin, the pellet was resuspended in 1 mL 10% glycerol, aliquoted to 50 μL, flash-frozen in liquid nitrogen and stored at −80 °C. *E. coli* electrocompetent cells were generated using a previously published method [30].

Electroporation was adapted from [29]. Briefly, competent cells in 50 μL aliquots were thawed on ice and 1–2 μL plasmid was added to the suspension and transferred to a chilled cuvette. The mixture was left on ice for 10 min before electroporation in an Eppendorf Electroporator 2510 (USA Scientific, Inc.—BioCT, Ocala, FL, USA) with a set voltage of 2500 V for methylotrophic cells and 1800 V for *E. coli* cells. Immediately following electroporation, 1 mL of NB media (methylotrophs) or 1 mL SOC media (*E. coli*) was added to the cuvette. Cultures were transferred to a microcentrifuge tube and then placed in a 28.5 °C incubator at 175 rpm for 20–24 h for methylotrophs. For *E. coli* strains, the tubes were placed in a 37 °C, 500 rpm shaker for 1–2 h. Following incubation, cells were added to their respective plates with appropriate antibiotics. Colonies were confirmed through PCR with the primers specific to the insert. All strains generated throughout the study are detailed in Table 2. 

### 2.3. Terpene Production and Extraction

A total of 4–5 colonies were inoculated in 3 mL MP with methanol (125 mM) and grown at 28.5 °C with shaking (175 rpm). After 3–4 days, the starter culture was used to inoculate a 50 mL culture in a 250 mL Erlenmeyer flask. Cultures were left to grow for 16 h or until an OD_600_ of 0.3–0.6 was reached. For the pAH1- and pAH4-containing strains, half of the cultures were given LaCl_3_ (2 μM) and all were returned to the shaker for 48 h. For strains possessing pAH2 and pAH3, the 50 mL cultures were induced by adding LaCl_3_ and returned to the shaker for 48 h. Overnight cultures of the *E. coli* strains were started from frozen glycerol stocks in 5 mL LB plus antibiotics and grown in a 37 °C, 200 rpm shaker. The next morning, 50 mL of TB plus antibiotics were inoculated with 1 mL of culture and returned to the shaker for 4–6 h. Once cultures reached an OD_600_ of 0.5–0.6, induction was performed by adding 40 mM sodium pyruvate, 2 mM MgCl_2_ and 1 mM IPTG. Cultures were incubated in a 20 °C, 200 rpm shaker for 48 h. 

Cultures were collected and ~20 mL hexane with 1 ng/μL Eicosene as internal standard was added. Cells were then sonicated using the bulk tip on a Misonix S4000 Ultrasonic Liquid Processor (Misonix, Inc., Farmingdale, NY, USA) with the following parameters: 60 amps, 100 s run, 10 s on and 5 s off. After sonication, 5 mL of 100% EtOH was added, and the flasks were then placed on a slow shaker at room temperature for 2 h. The top hexane layer was transferred to a glass tube and samples were reduced under a nitrogen stream to roughly 1.5 mL and transferred to a mass spectrometry glass vial. 

The lanthanide-dependent patchoulol formation was determined in triplicate, comparing growth with and without lanthanides and by using 100 mL cultures of *M. extorquens* strains AM1/pAH2 or CM502/pAH2 harboring *PcPatS* that were grown to an OD_600_ of 0.3–0.6 at 175 rpm shaking speed. Lanthanide (2 µM LaCl_3_) was provided for the plus lanthanide condition and a 20 mL dodecane overlay was added to all cultures, with further incubation at 30 °C for 3 days. The dodecane layer was removed and measured directly on gas chromatography with mass spectrometry (GC-MS) to detect patchoulol peaks.

### 2.4. GC-FID/GC-MS Analysis

GC-FID analysis was performed using an Agilent 7890A system (Agilent Technologies, Santa Clara, CA, USA) equipped with a 19091S-433 (30 m × 250 μm × 0.25 μm) column and chromatography with the following parameters: helium carrier, rates: 40 °C for 1 min, 40 °C/min hold for 2 min at 200 °C, 20 °C/min to 250 °C, 15 °C/min to 280 °C, 40 °C/min hold at 320 °C for 3 min, splitless: 250 °C, with an injection volume of 1 μL. The patchoulol synthase hexane samples were compared to a patchouli oil standard and the *E. coli*/pAH1 samples. Comparison of the controls and standard showed the retention time of patchoulol at 7.73 min. For the casbene samples, a previously collected plant extract and the *E. coli* samples were used as controls. The retention time for casbene was determined at 9.35 min. To further confirm the patchoulol and casbene peaks, representative samples were run on an Agilent A GC-MS instrument with a 30 m VF-5 column and the following parameters: helium carrier, rates: 40 °C hold for 1 min, 40 °C/min hold 200 °C for 4.5 min, 20 °C/min to 240 °C, 10 °C/min to 280 °C, 40 °C/min hold 320 °C for 3 min, splitless: 275 °C and injection volume of 1 μL. Statistical analysis of the GC-FID data was conducted through a one-way ANOVA test of the product to internal standard peak ratios. 

### 2.5. Competition Experiments

*Nicotiana benthamiana* plants were grown for four weeks in Sure-mix soil (Michigan Grower Products, Inc., Galesburg, MI, USA) in a growth room under controlled conditions with a 16-hour day cycle at 24 °C and an 8-hour night cycle at 17 °C. Methylotrophic cultures of CM502, AM1/pES503, CM502/pAH1, CM502/pAH2, CM502/pAH3, CM502/pAH4 were started in 3 mL MP media with methanol and tetracycline when appropriate. After 4 days, 500 mL of MP media with methanol (125 mM), 2 µM LaCl_3_ and tetracycline were inoculated with 1.5–2 mL of the starter cultures and were returned to the shaker for 2 days. Cultures were combined and pelleted by centrifugation at 4000× *g* for 15 min at 4 °C. After two washing steps, the final pellet was resuspended in ddH_2_O at half the initial volume, and the OD_600_ of each sample was determined. For coculture inoculation, resuspended cells were mixed in a 1:1 ratio at an OD_600_ of 1 each. Plants in replicates of three were either left unmanipulated or inoculated with CM502 alone, AM1/pES503 alone, AM1/pES503:CM502/pAH1, AM1/pES503:CM502/pAH2, AM1/pES503:CM502/pAH3 or AM1/pES503:CM502/pAH4. For this, plants were inverted, and the leaves were dipped into the bacterial cultures until all were fully coated. After inoculation, plants were returned to the growth room for 4 days. Bacteria were re-isolated using two medium-sized leaves from each plant. Three 12 mm leaf disks were placed in a sterile 50 mL falcon tube, and 50 mL 100 mM phosphate buffer was added. Tubes were shaken at 22 °C at 120 rpm for 10 min. A total of 100 μL of buffer for the single strain inoculations and 60 μL of buffer for the double strain inoculation plants was plated on MP agar with methanol and LaCl_3_ and MP agar with methanol, LaCl_3_ and tetracycline. Plates were covered in tin foil and allowed to grow in the incubator for 5 days before counting CFUs for both pink and white colonies in the presence and absence of tetracycline. 

## 3. Results and Discussion

Patchoulol synthase under a constitutive promoter fails to produce patchoulol

The pAH1 vector (encoding PcPatS under the constitutive promoter *mTac*) was electroporated into strain AM1, CM502, and grown until the exponential phase. At this point, LaCl_3_ was added to half of the methylotrophic cultures. After 48 h of expression, the cultures were harvested. For the *E. coli* controls, the patchoulol synthase was cloned in pET-28b(+) (EMD Millipore, Burlington, MA, USA) and transformed into chemically competent OverExpress^TM^ C41(DE3) cells (Lucigen, Middleton, WI, USA). Expression was performed as described previously [31]. The extractions of patchoulol with hexane were obtained from each culture and run on a GC-FID instrument. No patchoulol was detected in any sample. It is possible that the strength of the *mTac* promoter is insufficient for an efficient production of patchoulol that is above the detection limit of our instrument. This is consistent with *mTac* being only a moderate-strength promoter in both methylotrophic and *E. coli* systems [32]. It was recently suggested that the engineering of microbial terpenoid biosynthesis requires strong promoters to induce a metabolic pull toward the target product, regardless of the use of native, synthetic or inducible promoters [33]. The same study indicates an inherent benefit from overexpression of potential rate-limiting enzymes to increase the supply of the five-carbon precursors or the isoprenyl diphosphate synthase affording the direct precursor farnesyl diphosphate with synergistic effects of deploying multiple genes [33].

Native methylotrophic promoter successfully produces patchoulol

To test an alternative promoter, a second construct with a native, lanthanide-inducible methylotrophic promoter was generated, pAH2. The same experimental procedures were conducted, except all the cultures were given LaCl_3_ to induce the promoter. Analysis of the extracts showed the system was producing patchoulol. In comparison to baseline methylotroph samples and standard control, the patchoulol peak at retention time 7.73 min was found in all the AM1/pAH2 and CM502/pAH2 samples (Figure 2). This demonstrates that the system can produce patchoulol and is further evidence *M. extorquens* is capable of sesquiterpene synthesis. The analysis of the product and the internal standard peak in each sample were conducted and used for calculating the relative yield (Figure 3). This analysis showed, under these conditions, no significant difference between the relative product collected from the AM1/pAH2 and CM502/pAH2 samples. The lack of difference between the pink and colorless cells may be attributed to the low levels of product and may indicate a bottleneck in the pathway. Additionally, it indicates the natural MEP precursors within the CM502 mutant are not generating an increased pool. As discussed above, overexpression of multiple genes upstream of the terpene synthase may provide a future strategy to increase the carbon flux toward patchoulol [33].

Impact of the lanthanide switch on production of patchoulol

Inducible systems are highly attractive in metabolic engineering. To further characterize the use of the *xox1* promoter as a lanthanide switch, we tested whether patchoulol biosynthesis in *M. extorquens* is induced by the presence of LaCl_3_. Both AM1/pAH2 and CM502/pAH2 were tested for patchoulol production with or without the addition of LaCl_3_ (2 µM) (Figure 4). In both strains, patchoulol is only seen in the presence of LaCl_3_, indicating induction of *PcPatS* gene expression by the *xox1* promoter. This supports the *xox1* promoter acting as a lanthanide switch that enables inducible expression of biosynthetic pathways. Future studies could further characterize this lanthanide switch, investigating the sensitivity to different LaCl_3_ concentrations. Similarly, future investigation into the *xox1* switch in both AM1 and CM502 may quantitatively elucidate the terpenoid biosynthetic capacity of each strain. In the experiments presented here, both strains may still be limited by *PcPatS* production, and future improvements to *PcPatS* expression with *xox1* (for example, higher LaCl_3_ concentrations) may enable greater patchoulol production. Here, it is clear that patchoulol production in *M. extorquens* is induced by the addition of LaCl_3_ when using the *xox1* promoter, and there is an opportunity to further improve this system in the future.

Assessing the methylotrophic engineering of diterpenes

A diterpene pathway using the *mTac* promoter was generated in the AM1 strain to test the availability of the later C_20_ terpene precursor, GGDP. To our understanding, there is no report of diterpene production within *M. extorquens* AM1. The gene encoding casbene synthase was cloned under the *xox1* promoter, and no casbene was detected in the methylotrophic samples (2 µM LaCl_3_). The lack of product is potentially influenced by the availability of the GGDP precursor. Insufficient carbon flow through the pathway to GGDP or enzyme activity of GGDP synthase may be contributing factors. 

Methylotrophic and *E. coli* strains harboring the pAH4 constructs with the *mTac* promoter were generated to test diterpene synthase expression. The resulting chromatograms from the *E. coli* and standard hexane samples (Appendix A) show the *E. coli* control is producing both the terpene precursor at retention time 9.035 min (1) and the final diterpene product, casbene, at retention time 9.35 min (2). The methylotrophic samples showed no detectable casbene, possibly due to moderate expression by the constitutive promoter.

Methylotrophic engineered strains can compete for colonization in the phyllosphere

*M. extorquens* is a known colonizer of the phyllosphere and metabolizes the methanol released from plant catabolism of pectin. The success of engineering a methylotrophic sesquiterpene system provides the opportunity for creating a production platform on plant leaf surfaces. To determine if the engineered strain can compete with other methylotrophic strains for colonization, a competition experiment in the phyllosphere was conducted. The engineered strain was mixed with a control strain and used to inoculate plant leaves. Strains were re-isolated by selecting with methanol and LaCl_3_. The resulting plates of the recovered strains show a difference between the CM502/pAH2 and CM502/pAH3 conditions (Figure 5). Specifically, the CM502/pAH2 strain was recovered in low numbers compared to the control AM1/pES503 strain. In contrast, for the CM502/pAH3 strain, the recovered bacteria show a far higher proportion of white colonies, indicating it out-competed the AM1/pES503 control strain. The differences in recovery are possibly associated with the burden of terpene production. However, all engineered strains were recovered. A current established system for terpene expression within plant leaves uses transient infection through agrobacterium [31]. A combination of this expression system and the methylotrophic expression on the leaf surface holds the potential for increasing terpene products generated from one individual plant. 

## 4. Conclusions

This study adds further evidence that the methylotrophic strain of *M. extorquens* AM1 has a native, albeit low, capability for production of the C_15_ sesquiterpene patchoulol. In contrast, the C_20_ diterpene product casbene failed to accumulate at detectable levels, which could indicate limiting availability of the precursor, geranylgeranyl diphosphate. Here, we emphasize engineering of *M. extorquens* using a lanthanide-dependent native promoter. This promoter is a unique addition to the growing number of highly specific inducible systems, critical for synthetic biology applications. Comparison of a carotene-free strain with the wildtype strain did not result in a higher yield, indicating that the limitation may reside either in the recombinant heterologous activity of the terpene synthase or the endogenous precursor pathway and lack of C_5_ precursor building block availability. Lastly, the strains generated were inoculated in the phyllosphere and shown to colonize the model plant *Nicotiana benthamiana* in sufficient capacity to be recovered from leaves under competitive conditions. 

## Figures and Tables

**Figure 1 microorganisms-12-00500-f001:**
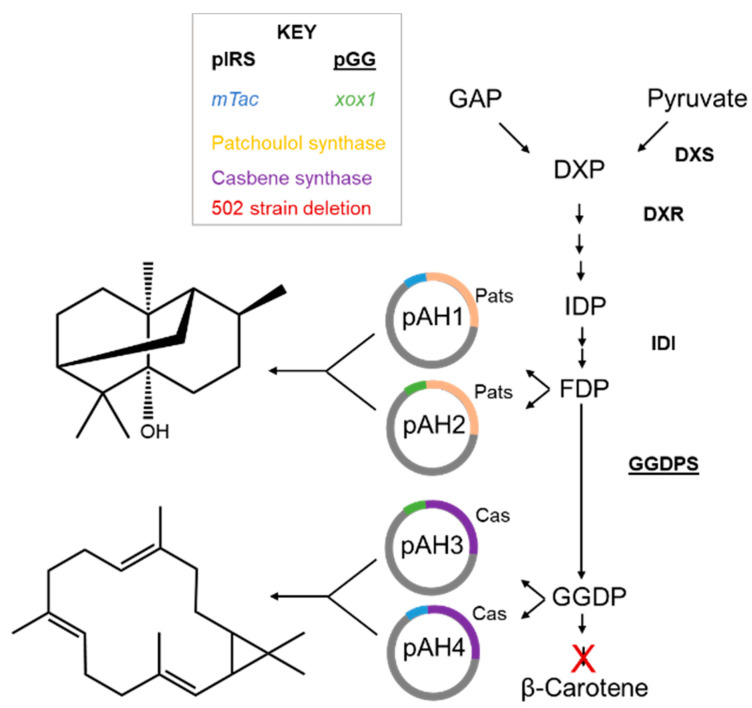
Scheme of vector insertion into MEP pathway. The four plasmids generated in the study (pAH1, pAH2, pAH3 and pAH4), as well as the previously established *E. coli* plasmids (pIRS and pGG), are shown. Pats, patchoulol synthase; Cas, casbene synthase.

**Figure 2 microorganisms-12-00500-f002:**
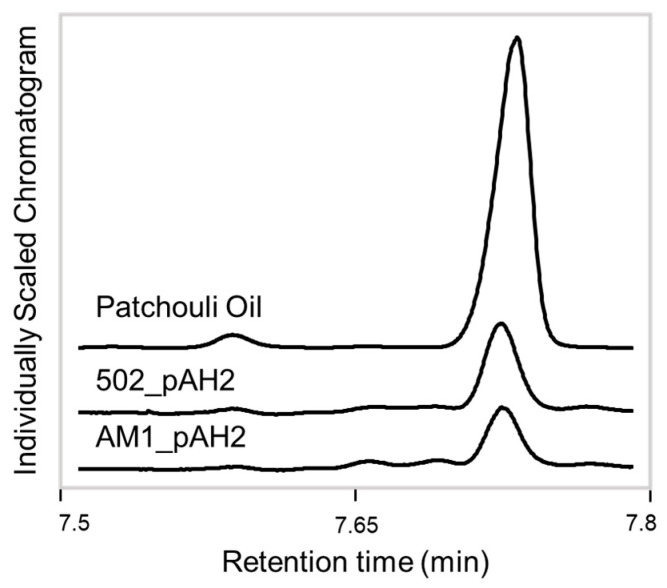
GC-FID chromatogram of patchoulol yields. Total ion chromatograms focused on the patchoulol peak from the patchouli oil standard, CM502_pAH2 and AM1_pAH2 samples. The peak retention time of patchoulol was determined at 7.73 min and is present in all three samples at varying intensities.

**Figure 3 microorganisms-12-00500-f003:**
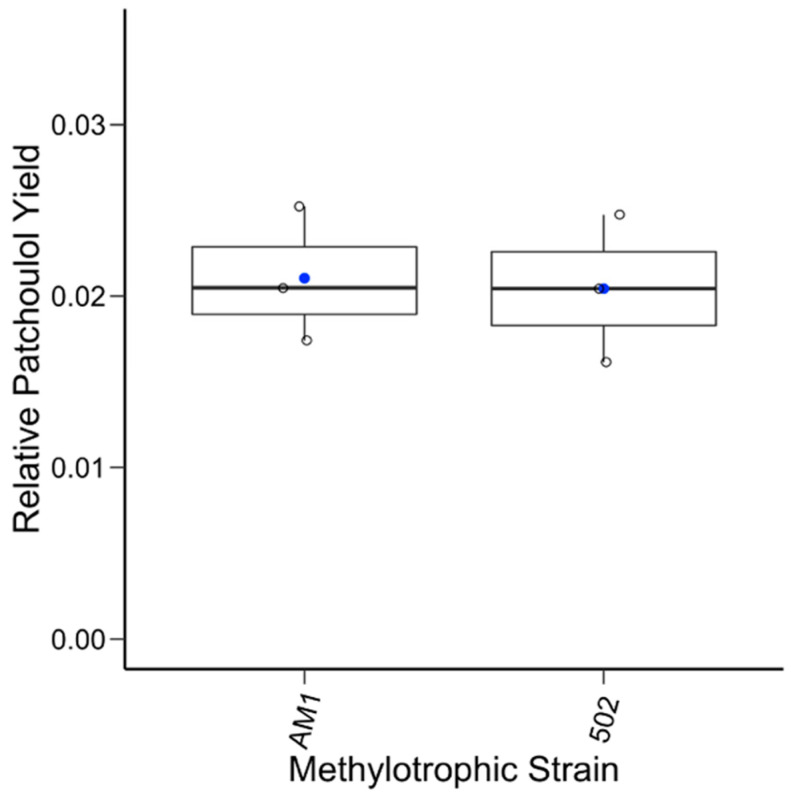
Relative patchoulol yields from the native promoter. The relative yield of patchoulol for each sample (open circle) from the two methylotrophic strains is plotted along with the mean (blue circle). A one-way ANOVA test was conducted, and no statistical significance between the two sample sets was found.

**Figure 4 microorganisms-12-00500-f004:**
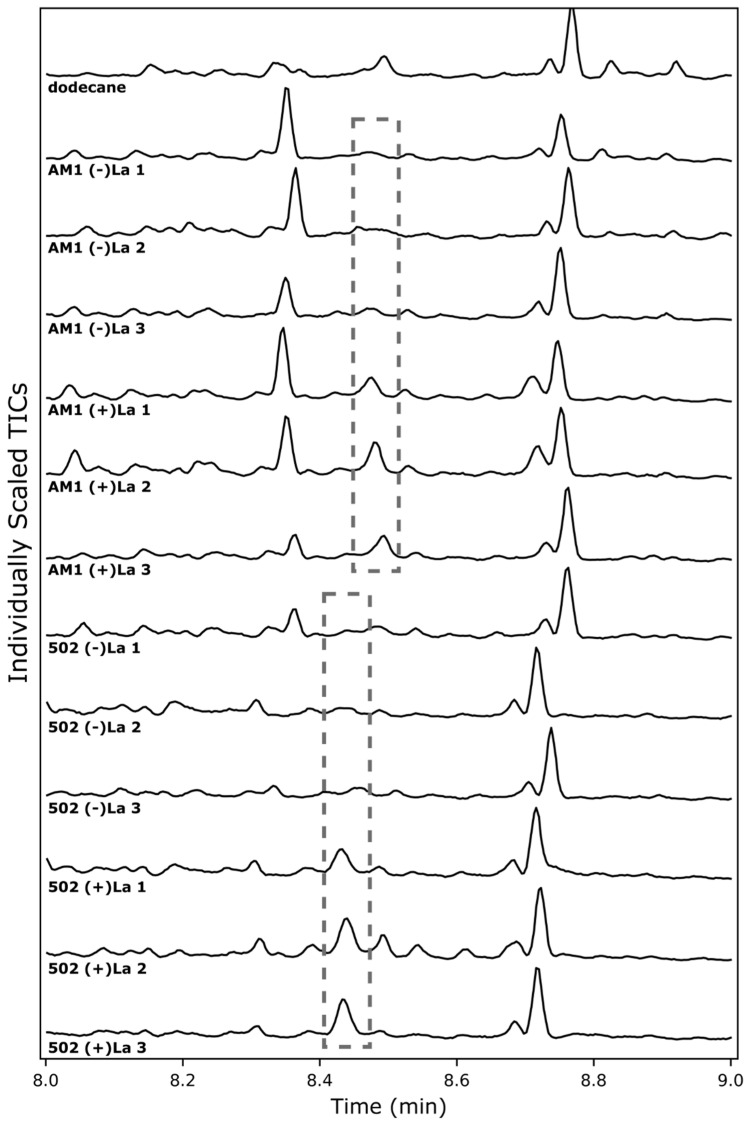
Chromatograms of M. extorquens pAH2 cultures with or without induction by LaCL3. Dashed boxes indicate elution of patchoulol from GC-MS analysis. Triplicate samples are individually displayed and numbered 1, 2 or 3, with the lack of (-) or presence of (+) LaCl_3_.

**Figure 5 microorganisms-12-00500-f005:**
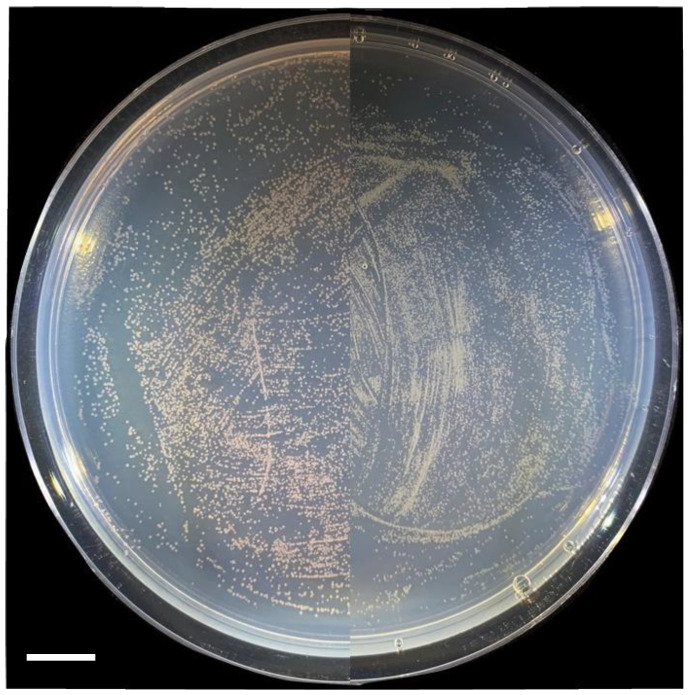
Recovered bacteria exhibit differences in phenotype. The phyllosphere competition produces a clear color difference in the bacterial cells recovered from the leaf surface. The CM502/pAH2 strain is found in low numbers (**left**). The CM502/pAH3 strain dominates the pink control strain AM1/pES503 (**right**). Scale bar, 1 cm.

**Table 1 microorganisms-12-00500-t001:** Oligonucleotides used in the study.

Primer	Description	Target Gene
1-pAP5.for	CGTTCCTGACAACGAGCCTCCTT	pAP5 linearization
2-pAP5.rev	GCAGGCATGCAAGCTTGGCGTAA	pAP5 linearization
3-mtac.pats.lin.rev	GAAGATCTGAATTCGAGATGGAGTTGTATGCCCAAAG	Patchoulol synthase
4-Pats.pAP5.rev	GCCAAGCTTGCATGCCTGCTTAATATGGAACAGGGTGAAGGTACAACTGC	Patchoulol synthase
5-mTac.pAP5.for	GAGGCTCGTTGTCAGGAACGAAGAAATCTGAAATGAGCTGTTGACAATTA	*mTac* promoter
6-mTac.pAP5.rev	CTCGAATTCAGATCTTCGGG	*mTac* promoter
7-XoxF.pAP5.for	CGAATTCACTGGCCGTCGTTTTACA	pES503 linearization
8-Pats.xoxf.for	AACGACGGCCAGTGAATTCGATGGAGTTGTATGCCCAAAGT	Patchoulol synthase
9-DgTPS1.pAP5.for	CCCGAAGATCTGAATTCGAGATGGCTGCTGCTGTGTCCGAGTT	Casbene synthase
10-DgTPS1.pAP5.rev	CGCCAAGCTTGCATGCCTGCTCATCGGTTATAAGGAATTGGGTGGACGAA	Casbene synthase
11-DgTPS1.xoxf.for	AACGACGGCCAGTGAATTCGATGGCTGCTGCTGTGTCCGAGTT	Casbene synthase
12-venus.check.for	CGAGTCAGTGAGCGAGGAA	Sequencing check
13-venus.check.rev	CTACTTCACTGTTGGGGCCG	Sequencing check

**Table 2 microorganisms-12-00500-t002:** Plasmids and strains used in the study.

Strain or Plasmid	Relevant Trait(s)	Source
pAP5	pCM62 promoterless venus	Skovran et al. [26]
pES503	pAP5 with *pxox1*	Sonntag et al. [16]
pAH1	pAP5 ∆venus_*pmTac*_*PcPatS*	This study, derived from pAP5, Skovran et al. [26]
pAH2	pES503 ∆venus_*pxox1*_*PcPatS*	This study, derived from Sonntag et al. [16]
pAH3	pES503 ∆venus_*pxox1_DgTPS1*	This study, derived from Sonntag et al. [16]
pAH4	pAP5 ∆venus_*pmTac*_*DgTPS1*	This study, derived from Sonntag et al. [16]
pIRS	*DXS*, *DXR*, *IDI*	Morrone et al. [12]
pGG	pACYCDUet with rAgGGPS	Cyr et al. [28]
*E. coli*_pIRS_pGG_pAH1 (strain)	*E. coli* with *pmTac_PcPatS*	This study, derived from Morrone et al. and Cyr et al. [12,28]
*E. coli*_pIRS_pGG_pAH4 (strain)	*E. coli* with *pmTac_DgTPS1*	This study
AM1_pES503 (strain)	*M. extorquens* AM1 with *pxox1_venus*	This study
AM1_pAH1 (strain)	*M. extorquens* AM1 with *pmTac_PcPatS*	This study
CM502_pAH1 (strain)	*M. extorquens* CM502 with *pmTac_PcPatS*	This study
AM1_pAH2 (strain)	*M. extorquens* AM1 with *pxox1_PcPatS*	This study
CM502_pAH2 (strain)	*M. extorquens* CM502 with *pxox1_PcPatS*	This study
AM1_pAH3 (strain)	*M. extorquens* AM1 with *pxox1_DgTPS1*	This study
CM502_pAH3 (strain)	*M. extorquens* CM502 with *pxox1_DgTPS1*	This study
AM1_pAH4 (strain)	*M. extorquens* AM1 with *pmTac_DgTPS1*	This study
CM502_pAH4 (strain)	*M. extorquens* CM502 with *pmTac_DgTPS1*	This study

*PcPatS*: Patchoulol synthase; *DgTPS1*: Casbene synthase.

## Data Availability

Data are contained within the article and Appendix A.

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
