# Peer review of "Engineering Terpene Production Pathways in Methylobacterium extorquens AM1"

_microorganisms, 2024, doi:10.3390/microorganisms12030500_

Round 1
Reviewer 1 Report
Comments and Suggestions for Authors
Dear Authors,
I read your article with great pleasure.
Terpenes and terpenoids naturally occur in plants and essential oils and are introduced as key ingredients in the design and production of novel biologically active compounds. Terpenoids are aromatic principles of several plants and are interesting to researchers primarily for formulating fragrances for perfumes. For example Patchoulol is a sesquiterpene alcohol found in patchouli. Patchouli oil is an important material in perfumery. The (−)-optical isomer is one of the organic compounds responsible for the typical patchouli scent. Patchoulol is also used in the synthesis of the chemotherapy drug Taxol. Isolating an individual terpene from an essential oil is quite difficult. It is no less difficult to synthesize these compounds due to the complexity of their structure and the tendency to skeletal rearrangements. In this regard, the presented work is extremely relevant. Due to the fact that terpenes are very labile and have a high tendency to rearrange, it seems to me that the you should pay more attention to proving the structure of the two obtained terpenoids. The peak retention time is good parameter, but insufficient in the case of terpenes… In my opinion, it is necessary to provide at least the mass spectra of the resulting compounds, as well as determine the angle of rotation.
What does it mean letter “s” in Pats (Fig. 1)? It is better to explain what (1) means (not only (2)) in the caption under Figure 5.
Sincerely, Reviewer
Author Response
Response to reviewer’s comments and suggestions
Reviewer 1
Terpenes and terpenoids naturally occur in plants and essential oils and are introduced as key ingredients in the design and production of novel biologically active compounds. Terpenoids are aromatic principles of several plants and are interesting to researchers primarily for formulating fragrances for perfumes. For example Patchoulol is a sesquiterpene alcohol found in patchouli. Patchouli oil is an important material in perfumery. The (−)-optical isomer is one of the organic compounds responsible for the typical patchouli scent. Patchoulol is also used in the synthesis of the chemotherapy drug Taxol. Isolating an individual terpene from an essential oil is quite difficult. It is no less difficult to synthesize these compounds due to the complexity of their structure and the tendency to skeletal rearrangements. In this regard, the presented work is extremely relevant. Due to the fact that terpenes are very labile and have a high tendency to rearrange, it seems to me that the you should pay more attention to proving the structure of the two obtained terpenoids. The peak retention time is good parameter, but insufficient in the case of terpenes… In my opinion, it is necessary to provide at least the mass spectra of the resulting compounds, as well as determine the angle of rotation.
What does it mean letter “s” in Pats (Fig. 1)? It is better to explain what (1) means (not only (2)) in the caption under Figure 5.
Sincerely, Reviewe
Author response: We thank the reviewer for the positive response and appreciate the concerns regarding the stability of terpenes. We have indeed had examples of unstable products during our research activities. However, with respect to the structure of the products we would like to state that (i) rearrangement of patchoulol requires harsh conditions (e.g., treatment with lead acetate, or peroxy acid), both leading to fragmentation or significant structural changes; (ii) our authentic analytical standard is stable; (iii) the corresponding enzyme patchoulol synthase is used as an authentic reference enzyme. We added the corresponding reference (Luo et al., 2016). For casbene we do not have an authentic standard but use a well-established reference enzyme. We added the corresponding reference (Bibik et al., 2022). As for the angle of rotation, we could not obtain sufficient product from the methylotrophic strains for purification and further analysis. Before scaling this system, we believe it would be rational to improve the production capacity, as described in the revised text.
Thank you. The ”s” refers to the enzyme product Patchoulol Synthase (accession number: AY508730). We have this now clarified for both enzymes in the legend.
Thank you for catching the non-described compound (1) in the legend of previous figure 5 (now supplemental figure 1). We added “(1), Geranylgeraniol, product of unspecific phosphatase activity from geranylgeranyl diphosphate.”

Reviewer 2 Report
Comments and Suggestions for Authors
The paper presented by Hurt et al. aimed to presentation of way to production of terpenes in Methylobacterium extorquens AM1. Authors underlined significance of the secondary plant metabolites in various branches of industry along with complications during their synthesis. On the whole, the paper is well prepared. Authors descibed in detail methods used for the aim of paper. Results are presented in form of a few figures along with their detail descirption. Issue of the paper is suitable for the Journal but some points have to be improved/corrected:
Abstract: too long. Please shorten the introduction of the abstract and include more information about obtained results. Simultaneously, please shorthen whole section.
Discussion: plase provide more detail discussion. Results are well presented, but discussion is insufficient. Please provide suitable up-to-date references.
Conclusions: why this part include table? The table should be included in discussion.
Author Response
Reviewer 2
Comments and Suggestions for Authors
The paper presented by Hurt et al. aimed to presentation of way to production of terpenes in Methylobacterium extorquens AM1. Authors underlined significance of the secondary plant metabolites in various branches of industry along with complications during their synthesis. On the whole, the paper is well prepared. Authors descibed in detail methods used for the aim of paper. Results are presented in form of a few figures along with their detail descirption. Issue of the paper is suitable for the Journal but some points have to be improved/corrected:
Abstract: too long. Please shorten the introduction of the abstract and include more information about obtained results. Simultaneously, please shorthen whole section.
Author response: The abstract was streamlined as is now at 287 words. Additional information for specific results was added.
Discussion: plase provide more detail discussion. Results are well presented, but discussion is insufficient. Please provide suitable up-to-date references.
Author response: Details and specifically a description of other systems and the impact of both strong promoters as well as engineering of the precursor pathway were added. A recent relevant study is now cited.
Conclusions: why this part include table? The table should be included in discussion.
Author response: Thank you. The table is now more clearly labelled and can be found after the references.

Reviewer 3 Report
Comments and Suggestions for Authors
Attach file

Attach file
Author Response
Reviewer 3
Open Review
|
Are the conclusions supported by the results? |
( ) |
(x) |
( ) |
( ) |
Comments on the Quality of English Language
Attach file
Submission Date
23 January 2024
Date of this review
31 Jan 2024 01:43:14
Author response: We carefully read the manuscript and made edits, also in accordance with the comments of the other reviewers.
The manuscript ID microorganisms-2863394 is entitled: “Engineering Terpene
Production Pathways in Methylobacterium extorquens AM1”.
The authors describe terpene production by different strains of Methylobacterium
extorquens as alternative that metabolize renewable and affordable carbon sources may offer
unique sustainable opportunities to produce terpenes.
The authors focus is on two terpene products (patchoulol and casbene) because have
relevant application in the perfume and cosmetic industry.
Figures and Scheme are in good quality and with adequate information for justified the
production and characterization of the structures and use of strains of bacteria in this study.
In conclusion: The results of summary the terpene pathway of microbial strains. However, the
conclusion must be rewritten.
Author response: We have re-written and re-phrased statements in the conclusion section and clarified our findings, including key negative results of the engineering attempts.
The manuscript is suitable for publication in Microorganisms
Major corrections:
p.5, line 161: …for 15 min at 4°C. …. Change by … for 15 min at 4 oC….
Check °C throughout the manuscript
Author response: The National Renewable Energy Laboratory, a national laboratory of the U.S. Department of Energy, editorial standard states “Use a degree symbol (°) with temperatures expressed in the Celsius and Fahrenheit scales but not with kelvins (just use K). Don't leave a space between the number and the letter for °C and °F, but leave a space between the number and K.”. As our research is funded by the DOE, we prefer to adhere to their guidelines.
- 11, line 348-349: In the conclusion the sentence could be deleted “Previous research on
methylotrophic engineering of terpene pathways has utilized the sesquiterpene, α-humulene
[13,30].”
Author response: The sentence was removed.
- 11-12, line 362: Final of conclusion: This table that it contains the primer, description and
target genes was not cited in the text and is out of place. In my opinion this is not conclusion.
Author response: (See also response to reviewer 1) We carefully checked the main text, removed one instance of table 1 and ensured both tables are cited at the corresponding section in the material and methods. The table is now more clearly labelled and can be found after the references.
p.12, line 384: …metabolically engineered Corynebacterium glutamicum… Change by ….
metabolically engineered Corynebacterium glutamicum…
Check scientific names (in italic) throughout the manuscript
Author response: We thank the reviewer for catching this oversight. We corrected 26 instances of scientific names throughout.

Reviewer 4 Report
Comments and Suggestions for Authors
1. The abstract should be rewritten with more clearer results
2. the diterpene casbene should be the casbene diterpene in all the manuscript
3. Table 2. Plasmids and strains used in the study. this table should be clarified more especially the plasimids codes
4. Figure 2. GC-FID chromatogram of patchoulol yields. The authors should insert the main GC-FID chromatogram as supporting data
5. Figure 4. Chromatograms of M. extorquens pAH2 cultures with or without induction by LaCL3. The authors should insert the main GC-FID chromatograms as supporting data
6. Figure 5. GC-FID confirmation of casbene production. The authors should insert the main GC-FID chromatograms as supporting data
7. Here, we focused.... possessive pronouns should be deleted from all manuscript
Comments on the Quality of English Language
Ok
Author Response
Reviewer 4
Open Review
(x) I would not like to sign my review report
( ) I would like to sign my review report
Quality of English Language
( ) I am not qualified to assess the quality of English in this paper
( ) English very difficult to understand/incomprehensible
( ) Extensive editing of English language required
( ) Moderate editing of English language required
(x) Minor editing of English language required
( ) English language fine. No issues detected
|
Yes |
Can be improved |
Must be improved |
Not applicable |
|
|
Does the introduction provide sufficient background and include all relevant references? |
(x) |
( ) |
( ) |
( ) |
|
Are all the cited references relevant to the research? |
(x) |
( ) |
( ) |
( ) |
|
Is the research design appropriate? |
( ) |
(x) |
( ) |
( ) |
|
Are the methods adequately described? |
( ) |
(x) |
( ) |
( ) |
|
Are the results clearly presented? |
(x) |
( ) |
( ) |
( ) |
|
Are the conclusions supported by the results? |
(x) |
( ) |
( ) |
( ) |
Comments and Suggestions for Authors
- The abstract should be rewritten with more clearer results
Author response: (See also response to reviewer 1) The abstract was re-written, including additional details on results. We hope that it now reads more clear.
- the diterpene casbene should be the casbene diterpene in all the manuscript
Author response: Dating back to the early discovery in 1984 (doi: 10.1104/pp.81.2.335) the field has adopted “diterpene casbene”. Accordingly, our and other research yields about 1240 hits in google. We prefer keeping “diterpene casbene”.
- Table 2. Plasmids and strains used in the study. this table should be clarified more especially the plasimids codes
Author response: We clarified strains and origin of key parts in plasmids in Table 2, added four occurrences to the table in the corresponding material and methods section and hope that this context is now clearer.
- Figure 2. GC-FID chromatogram of patchoulol yields. The authors should insert the main GC-FID chromatogram as supporting data
- Figure 4. Chromatograms of M. extorquens pAH2 cultures with or without induction by LaCL3. The authors should insert the main GC-FID chromatograms as supporting data
- Figure 5. GC-FID confirmation of casbene production. The authors should insert the main GC-FID chromatograms as supporting data
Author response: We agree that the negative results of the lack of casbene production in M. extorquens does not strengthen the main text. We provide this now as Supplemental Figure 1, with the main text accordingly corrected. For figures 2 and 4, we prefer to keep these in the main text as they are key results and center stones of the narrative (successful production of patchoulol as well as the impact of the inducible lanthanide switch.
- Here, we focused.... possessive pronouns should be deleted from all manuscript
Author response: Two instances of possessive pronouns in the document are “(…) patchoulol that is above the detection limit of our instrument.” and “To our understanding, there is no report (…)”. We prefer to keep those pronouns as the wording would otherwise generalize context we believe is important.
Comments on the Quality of English Language
Ok
Submission Date
23 January 2024
Date of this review
30 Jan 2024 12:49:53

Round 2
Reviewer 4 Report
Comments and Suggestions for Authors
Accepted in the present form
Comments on the Quality of English LanguageOk